

# Manx shearwater (*Puffinus puffinus*) rafting behaviour revealed by GPS tracking and behavioural observations

Cerren Richards[1,2], Oliver Padget[3], Tim Guilford[3] and Amanda E. Bates[1,2]

[1] Department of Ocean Sciences, Memorial University of Newfoundland, St John's, Newfoundland and Labrador, Canada
[2] Department of Ocean and Earth Sciences, University of Southampton, Southampton, United Kingdom
[3] Oxford Navigation Group, Department of Zoology, University of Oxford, Oxford, United Kingdom

Corresponding author
Cerren Richards,
cerrenrichards@gmail.com,
cerridwenr@mun.ca

## ABSTRACT

Before visiting or leaving their remote island colonies, seabirds often engage in a behaviour termed 'rafting', where birds sit, often in groups, on the water close to the colony. Despite rafting being a widespread behaviour across many seabird taxa, the functional significance of rafting remains unknown. Here we combine global positioning system (GPS) tracks, observational and wind condition data to investigate correlates of rafting behaviour in Manx shearwaters (*Puffinus puffinus*) at a large colony on Skomer Island, Wales. We test (1) the influence of wind direction on rafting location and (2) whether raft size changes with respect to wind speed. Our approach further allows us to describe day-night trends in (3) raft distance from shore through time; (4) the number of birds present in the nearshore waters through time; and (5) spatial patterns of Manx shearwater rafts in marine waters adjacent to the breeding colony. We find no evidence that wind direction, for our study period, influences Manx shearwater rafting location, yet raft size marginally increases on windier days. We further find rafting birds closer to the shore at night than during the day. Thus, before sunset, birds form a "halo" around Skomer Island, but this halo disappears during the night as more individuals return from foraging trips and raft nearer the colony on Skomer Island. The halo pattern reforms before sunrise as rafts move away from land and birds leave for foraging. Our results suggest that wind conditions may not be as ecologically significant for rafting locations as previously suspected, but rafting behaviour may be especially important for avoiding predators and cleaning feathers.

## INTRODUCTION

There is a wealth of information on the population biology of seabirds. This may be because seabirds are easy to observe—thousands to millions of individuals gather at single colonies, typically on isolated land masses, during the breeding season and can be monitored. In addition, due to strong site fidelity during breeding and chick rearing, breeding birds can be tagged with a variety of biologging devices which can provide detailed information on movement patterns through day-night cycles and can be successfully used to infer flying,

foraging and resting behaviours (*Votier et al., 2010*; *Dean et al., 2012*; *Carter et al., 2016*; *Yoda, 2019*). Thus, for practical reasons, seabird behavioural research activities focus on breeding colonies in spring and summer months. More recently, since individuals often return to the same nesting location in subsequent years, biologgers can be recovered following a wintering season to provide detail on migration routes and survivorship (*Guilford et al., 2009*; *Wakefield, Phillips & Matthiopoulos, 2009*; *Fifield et al., 2014*; *Fayet et al., 2017*).

In addition to relatively well-studied behaviours of seabirds during breeding seasons, many seabird species, such as tubenoses, sulids and auks, form conspicuous and dense aggregations on surface waters adjacent to breeding colonies, hereafter referred to as rafts. In general, temporary animal aggregations, when they are not directly the consequence of aggregated resources such as food, are thought to provide adaptive benefits to individuals through processes such as reduced personal predation risk, as proposed in the Selfish Herd Principle (*Hamilton, 1971*), increased vigilance or predation-risk assessment (*Magurran, 1990*), opportunities for social behaviour, or information exchange, as presented in the Information Centre Hypothesis (*Ward & Zahavi, 1973*). Recent technological gains have shed insight into the importance of seabird rafting behaviour for a number of species (*Wilson et al., 2009*; *Thiebault et al., 2014*; *Carter et al., 2016*). Social rafting can be associated with preening and cleaning behaviour. For example, Northern gannets (*Morus bassanus*) raft significantly more on outbound journeys compared to inbound journeys suggesting rafts are used to preen and clean feathers after spending periods at their colony (*Carter et al., 2016*). Rafting may also lead to greater foraging success (*Evans et al., 2016*), and information exchange through social communication (*Weimerskirch et al., 2010*; *Machovsky-Capuska et al., 2014*) such as reported for the Guanay cormorant (*Phalacrocorax bougainvillii*). This cormorant forms compass rafts during the day which indicate the location of food patches to conspecifics departing the colony (*Weimerskirch et al., 2010*).

While many causes of rafting have been proposed, the influence of weather on rafting presents a knowledge gap in our understanding of the behaviour (*Wilson et al., 2008*; *Carter et al., 2016*). Temporal variations in wind conditions may particularly affect rafting behaviour. Wind speed and direction may influence rafting by, for example, affecting how many birds are found in rafts versus foraging and other activities, and by influencing where birds form rafts relative to wind exposure. It is also unknown whether rafting behaviour changes through day-night cycles, yet biologging devices now offer the opportunity to study these potential changes. Manx shearwaters (*Puffinus puffinus*) raft closer to their breeding colonies as night progresses, and it is speculated to be associated with assessing predation threat on land (*Wilson et al., 2008*). The opposite pattern may be expected as dawn breaks if groups are using raft distances from shore to avoid predators. Studying these effects may give insight into the adaptive significance of rafting.

Here, we aim to identify key relationships between wind conditions and Manx shearwater rafting behaviour in nearshore waters surrounding their breeding colony across day-night cycles. We select the Manx shearwater, a small ca. 400 g tubenose seabird which forms rafts that are visible from land. With more than 316,000 breeding pairs, Skomer Island, UK,
has the largest colony of Manx shearwaters in the world (*Perrins et al., 2012*). Thousands of individuals raft together from late afternoon and remain in nearshore waters until dark when they visit their burrows to feed their chicks (*Brooke, 1990*; *Keitt, Tershy & Croll, 2004*; *Wilson et al., 2008*; *Freeman et al., 2013*). Due to the importance of the marine waters surrounding Skomer Island for rafting Manx shearwaters and other breeding seabirds, a 1,668 km$^2$ Special Protection Area (SPA) was classified in 2017 (*Natural Resources Wales, Joint Nature Conservation Committee, 2015*).

We combine a large global positioning system (GPS) dataset of Manx shearwater locations, observational data on raft size and location, and wind condition data over one month to test (1) the influence of wind direction on rafting location and (2) whether raft size changes with respect to wind speed. Our approach further allows us to describe day-night trends in (3) raft distance from shore through time; (4) the number of birds present in the nearshore waters through time; and (5) spatial patterns of Manx shearwater rafts in marine waters adjacent to the breeding colony. We predict that rafts will aggregate on the lee side of the island during high wind. While offshore, shearwaters prefer to fly in strong winds, around the colony they are motivated to raft and so may seek shelter on calmer water. Larger rafts may also be expected on windier days if their return from foraging is aided by high wind speeds. Further, rafting locations and patterns may change through day-night cycles if birds are rafting to avoid land-associated predation. If raft locations are determined in part by predator avoidance, we would expect to see rafts at a greater distance from shore during daylight hours when diurnal predators are active, for example gulls, but closer to shore during the night. This would allow groups to better assess predation threat and weather conditions at the colony before visiting their burrows. Additionally, if individuals are using rafts to clean feathers, we may expect to see prolonged rafting in nearshore waters before they enter their burrows at night and following burrow visits rather than direct departure to foraging grounds. We further expect to find more individuals near the colony during the night as they return from foraging trips to feed their chicks.

## METHODS

### Field methods: behavioural observations

We conducted field work on Skomer Island, a small island off the coast of West Wales, UK (51°44′08.9″N, 005°17′47.0″W). Behavioural observations were made of rafting Manx shearwaters surrounding the entire circumference of the island between August 5th and September 4th, 2016 from four observation stations (Skomer Head, South Haven, North Haven, Garland Stone; Fig. 1). These stations were selected because they avoided disturbing seabird burrows, offered full coverage of the island, and optimised rafting observations due to close proximity to the coastline and a clear field of view. The order of observation visits to each station was randomised to prevent systematic bias in the size of rafts and distance from shore due to the time of day. Behavioural observations were conducted by the same observer (Richards), which removed observer-related variability between estimates.

To record the size, bearing, and distance of each raft within a 2,000 m radius of the observation stations, each day, two hours before sunset, we used 8 × 42 binoculars, a
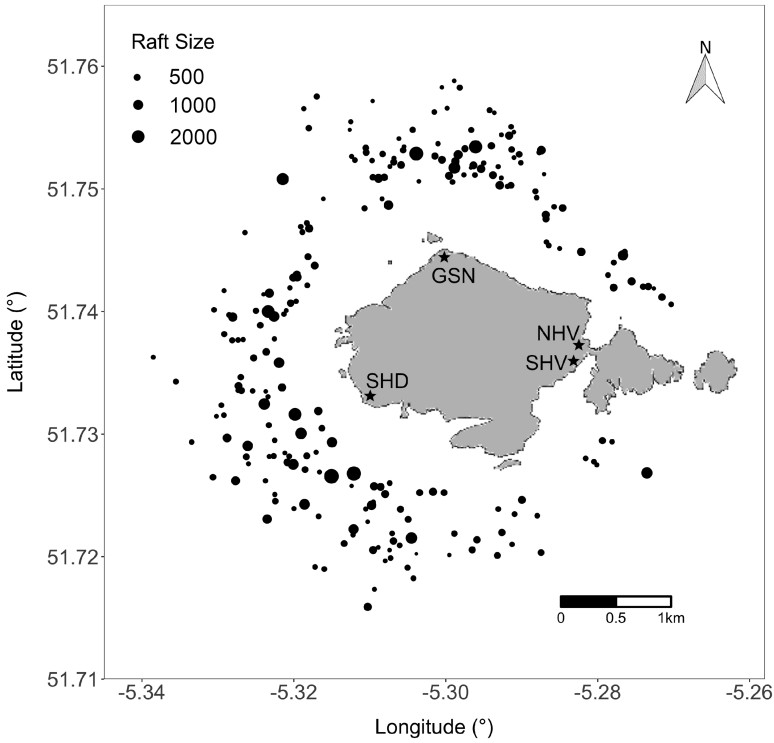

**Figure 1** **Location of rafting Manx shearwaters around Skomer Island two hours before sunset between August 5th and September 4th, 2016.** Each circle represents a raft and the size indicates the number of birds in each raft. Stars indicate the four sampling locations—Garland Stone (GSN), North Haven (NHV), Skomer Head (SHD) and South Haven (SHV). Map data © OpenStreetMap contributors. Map tiles © Stamen Design.

magnetic compass, and inclinometer. The inclinometer was calibrated to zero using a spirit level, and the tripod holding the inclinometer was made level at each observation station using an inbuilt spirit level. Rafts observed at the 2,000 m boundary and beyond were excluded in our analysis because we expected higher error associated with the inclinometer and counting individuals. Daily weather data were from a weather station at Wooltack Point (51°44′11.3″N, 005°14′50.6″W), 3,400 m away from the centre of Skomer Island. Rafting observations were not carried out on nine days during the study period due to compromised visibility resulting from heavy precipitation, fog, or wind speeds exceeding 18 ms$^{-1}$.

## Observational rafting distance and coordinates

The distance of each raft from the observation locations was calculated using the following basic trigonometric approach:

$$Raft\ distance = \frac{(C \pm T)}{D} \times H \qquad (1)$$

where $C$ = height of cliff at observation station (Garland Stone: 64.1 m; Skomer Head: 66.7 m; South Haven: 63.7 m; North Haven: 49.0 m), $T$ = tide height relative to mean

sea level, $D$ = declination distance, and $H$ = height of inclinometer (0.895 m). To determine the altitude and coordinates of the observation stations, values were taken from a GPS (Garmin etrex 10 software version 2.90). Tidal data were from Martin's Haven (51°43′59.9″N, 005°15′00.0″W), 3,200 m away from the centre of Skomer Island. The full trigonometric approach is detailed in Equation S1.

Spherical trigonometry was used to calculate the rafts coordinates along a great circle given the bearing and distance from the observation stations. For statistical analysis, the distance of each raft from the nearest point of Skomer Island's shoreline was estimated using the Haversine formula which calculates the great circle distance between two points.

### Field methods: GPS tracking

Between 31st May and 21st August, 2016, adult Manx shearwaters ($n = 83$) were fitted with GPS loggers (Mobile Action iGot-U gt120), housed in heat shrink plastic and secured to small bunches of contour feathers on the back using strips of marine Tesa® tape as in *Guilford et al. (2008)*. GPS loggers weighed 17 g, comprising approximately 4.25% of a Manx shearwater's typical body weight. GPS loggers were programmed to take location fixes every five minutes. Breeding birds were selected from specific burrows at a colony in North Haven on Skomer Island (51°44′14.0″N, 5°16′55.5″W). From each breeding pair, one bird was fitted with a GPS logger after its chick had been fed, or during partner changeovers for incubating birds. GPS loggers were removed the following night, or as soon as the individual returned to its burrow. Simultaneous deployments were avoided on pairs from the same nest. A maximum of 19 GPS loggers were simultaneously deployed during the study period. Handling time was kept to a minimum (normally less than 15 min). These GPS tracking methods have previously been shown not to have a measurable impact on breeding success at this study colony (*Dean et al., 2012*; *Shoji et al., 2015*).

This research was provided full approval by the University of Oxford Animal Welfare and Ethical Review Body and BTO Special Methods Technical Panel. Field experiments were approved by the Islands Conservation Advisory Committee (ICAC). A bird ringing permit was obtained from the British Trust for Ornithology (Permit Class and Number: / C / 5311).

### Identifying rafting from GPS data

GPS tracking can further extend our understanding of rafting behaviour during the night and further afield, which is a limitation of visual observations. Consequently, the study area was extended from 2,000 m to 5,000 m to capture a wider image of rafting behaviour, whilst remaining focused near the island. To ensure behaviour at the colony and on land were not included in the analysis, all GPS fixes on land were removed from the dataset.

The Vincente ellipsoid function was selected to calculate the geodesic distance between GPS fixes. For each bird, the speed between GPS fixes was calculated as distance change over time. Erroneous locations were removed from the GPS tracks by applying a speed filter to omit all data points that exceeded 30 ms$^{-1}$. Since rafting and flying can be distinguished based on speed, we employed an objective and repeatable method for choosing a speed threshold that identifies the speed that, above which, a bird is more likely to be flying and

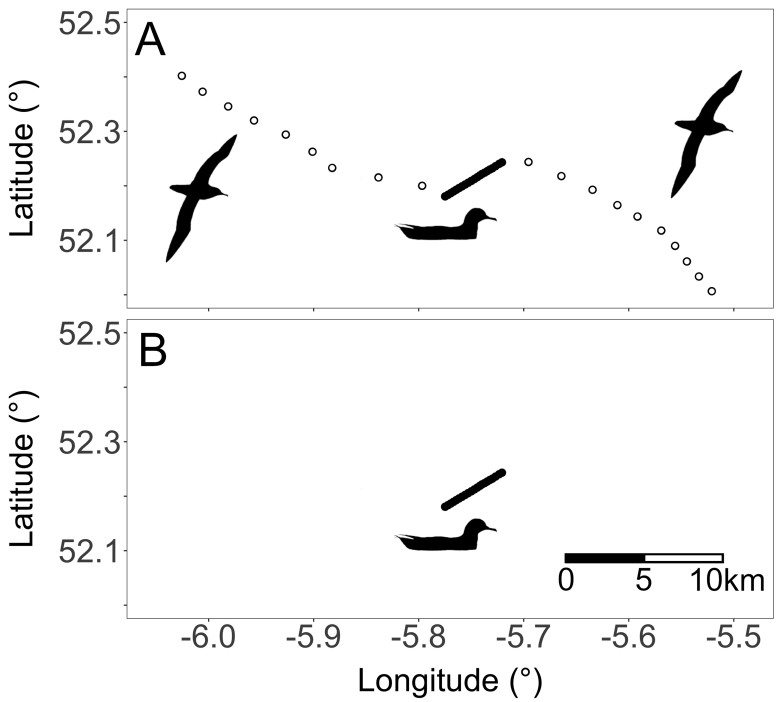

**Figure 2** **(A) Unfiltered and (B) filtered GPS fixes of a single Manx shearwater.** White dots indicate flying while black dots indicate rafting. Silhouettes represent the two behaviours within the GPS tracks. Behaviours are qualitatively identifiable because flying appears as single distanced GPS fixes whereas rafting is identifiable as close successive directional fixes (*Carter et al., 2016*).

below which, a bird is more likely to be rafting. To do this, we assumed that these two different behavioural states would each give rise to normally distributed speeds and thus, that the overall distribution of speeds would be a mixture of two normal distributions with a centre corresponding to each behavioural state. To parameterise these two normal distributions, we used an expectation maximisation algorithm, implemented using the 'mixtools' package in R and function *normalmixEM* with the number of components set at $k = 2$ (*Benaglia et al., 2009*), which iteratively fits two normal distributions to these emission data and finds the two means and standard deviations most likely to give rise to the observed data. Thus, from the 83 Manx shearwater GPS tracks, the speed at which a datum switched from being more likely to be attributed to the rafting centroid to being more likely to be attributed to flying was 2.5 ms$^{-1}$. Rafting was therefore defined as GPS fixes below a speed threshold of 2.5 ms$^{-1}$. All flying behaviour was filtered from the GPS tracks and only rafting behaviour was carried forward to further analyses. When mapped, the resulting rafting behaviours appear as close successive fixes showing a directional pattern following tidal or wind movements (*Dean et al., 2012*; *Carter et al., 2016*); Fig. 2).

## Mapping, data conversion, and analysis

All analyses and mapping were carried out in the R 3.5.0 programming language and environment (*R Core Team, 2018*). Maps and figures were created with the packages 'ggplot2' (*Wickham, 2016*) and 'ggmap' (*Kahle & Wickham, 2013*). Several data conversions

**Table 1** Coefficient values returned using a GLS modelling approach fitted with a temporal auto- correlation term of $q = 4$ to test for the relationship between Manx shearwater raft size and wind speed. Theta values are the autocorrelation terms for each time lag.

| AIC | BIC | logLik | | | | |
|---|---|---|---|---|---|---|
| 713.9 | 737.9 | $-349.9$ | | | | |

**Correlation structure: ARMA (0,4)**

**Parameter estimates**

| Theta 1 | Theta 2 | Theta 3 | Theta 4 | | | |
|---|---|---|---|---|---|---|
| 0.088 | 0.162 | 0.185 | 0.162 | | | |

| Predictor | Value | Std. Error | $t$-value | $p$-value | 2.5% CI | 97.5% CI |
|---|---|---|---|---|---|---|
| Intercept | 4.674 | 0.227 | 20.552 | 0.000 | 4.229 | 5.120 |
| Wind Speed | 0.065 | 0.031 | 2.087 | 0.038 | 0.004 | 0.126 |

**Standardized residuals**

| Min | Q1 | Med | Q3 | Max |
|---|---|---|---|---|
| $-3.118$ | $-0.597$ | $-0.035$ | 0.724 | 2.349 |

**Residual standard error**

1.146

| Total degrees of freedom | Residual degrees of freedom |
|---|---|
| 230 | 228 |

were required. To determine rafting direction, the bearing between each GPS fix was calculated using spherical trigonometry for each bird. All bearings were converted to radians and classified as circular data fitted to a Von Mises distribution. To explore the rafts' response to wind direction, the rafting time and date were matched to the nearest time and date of the weather data from Wooltack Point.

## Statistical tests

To test whether raft size changes with respect to wind speed within the behavioural observation data, a generalized least squares (GLS) model was constructed using the 'nlme' package and function *gls* (*Pinheiro et al., 2018*). Wind speed was included as a predictor (fixed effect), and raft size was log-transformed to meet the error structure assumption of homoscedasticity. Dates were converted to Julian days and temporal autocorrelation in raft size was modeled by fitting an ARMA process to the residuals. The number of auto-regressive parameters were specified as $p = 4$, and the ARMA was nested within each Julian day. Model results (based on the function *summary.gls* with default settings) are reported in Table 1, and the model structure and selection procedures are presented in the Code S1.

To test for a response in rafting direction (behavioural observations and GPS tracks) to wind direction, we used Jammalamadaka-Sarma circular correlation test using the package 'circular' and function *cor.circular* (*Agostinelli & Lund, 2017*). This test is a circular version of the Pearson's product moment correlation and was selected because it allows the calculation of a correlation coefficient using angular variables.

To describe the trends in (1) the number of birds in nearshore waters through time and (2) raft distance from shore through time, we fitted general additive mixed models
(GAMM) with Poisson distributions for count data. Bird ID was further included as a random effect for distance from shore through time. We used the package 'mgcv' and function *gamm* (*Wood, 2017*).

## RESULTS

### Rafting behaviour

From the behavioural observations, a total of 248 rafts were identified across 22 days, however, we removed 18 rafts because they fell outside the 2,000 m sampling boundary. Raft size was highly variable; the smallest raft consisted of 10 birds; the largest of 2,900 birds; and mean size was 336 ± 470 SD birds. A total of 77,390 individuals were counted over the duration of the study.

We found that Manx shearwaters rafted in all directions around Skomer Island (Figs. 1 and 3). Within the behavioural observation data, mean rafting distance from the island's shoreline was 924 ± 315 SD m, and no rafts were located closer than 276 m; the result was a halo effect around the island with an absence of rafts within a 300–500 m boundary two hours before sunset (Fig. 1). The GPS data also revealed that Manx shearwaters raft further from land during the day resulting in a halo effect around the island two hours before sunset and after sunrise (Figs. 3A–3C). However, the halo effect disappeared at night as tracked birds rafted closer to the island (Fig. 3D). Rafting accounted for 85%, and flying for 15% of GPS tracked Manx shearwater behaviour within the 5,000 m radius of Skomer Island. Around the colony, a significant proportion, 93%, of rafting occurred during the night whilst only 7% occurred during the day.

### Predictors of raft size

Wind speeds during the study period ranged from 2.1 ms$^{-1}$ to 17.6 ms$^{-1}$, and average wind speeds were 8 ms$^{-1}$. This is moderately representative of the season on Skomer Island because mean wind speed between April and September, 2016 was 7 ms$^{-1}$, and ranged from 0.3 ms$^{-1}$ to 20 ms$^{-1}$. More birds were present in rafts on windier days (Table 1). Residual autocorrelation was detected in the model of wind speed as a predictor of raft size. We accounted for this temporal autocorrelation with a lag of four days which significantly improved the model fit. When wind speeds were greater than 10 ms$^{-1}$, birds formed flocks of more than 10,000 individuals that failed to raft, or only settled for seconds. Birds were observed to be restless during periods of high wind, but were more settled in calm conditions.

### Predictors of rafting locations

Wind direction was variable throughout the study period, and came from all directions except the east. There was no effect of wind direction on the rafting location of Manx shearwaters around Skomer Island within the behavioural observation data (Jammalamadaka-Sarma Circular Correlation, correlation coefficient = −0.015, test statistic = -0.228, $p = 0.820$). For the GPS data, there was a small negative effect of wind direction on rafting direction, (Jammalamadaka-Sarma Circular Correlation, correlation coefficient = -0.065, test statistic = −3.20, $p = 0.001$). During the night, the vast majority

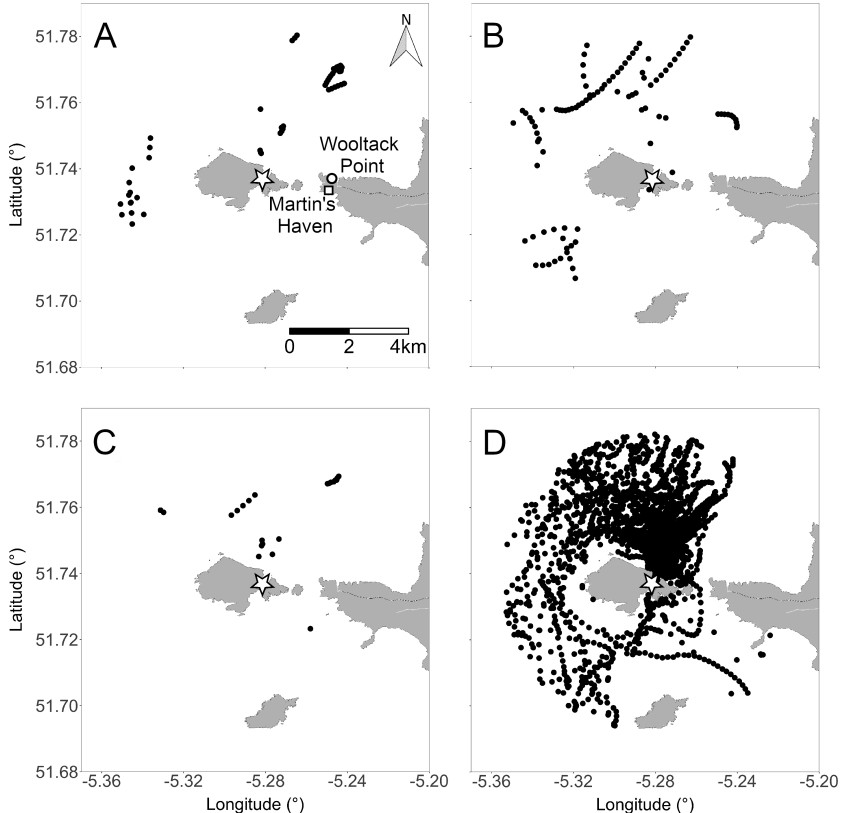

**Figure 3** **Rafting positions of 83 GPS-tracked Manx shearwaters within a 5,000 m radius of the colony across 45 days.** (A) Two hours after sunrise, (B) two hours before sunset, (C) during the day, and (D) during the night. Star indicates location of the North Haven colony. White circle indicates the location of the Wooltack Point weather station and white square indicates the location of Martin's Haven, the tidal predictions reference point. Rafting behaviour typically appears as close successive directional GPS fixes. Map data © OpenStreetMap contributors. Map tiles © Stamen Design.

of GPS tacked birds rafted adjacent to the North Haven colony, located in the Northeast of Skomer (Fig. 3D).

## Day-night trends in bird presence and rafting distance from shore

We found a non-linear trend in GPS tracked birds, where time of day affected the distance of rafts from the island and the number of birds in the nearshore waters. More birds arrived (Fig. 4A) and rafted closer to land (Fig. 4B) as the evening progressed, and as dawn approached birds moved further away (Fig. 4B) and left vicinity of the island (Fig. 4A).

## DISCUSSION

Rafting behaviour by seabirds is poorly documented and understood within the scientific community (*Sánchez-Román et al., 2019*). A number of studies have described distributions of rafts around colonies, but have not yet tested for relationships with possible environmental drivers (*Guilford et al., 2008*; *Wilson et al., 2008*; *Wilson et al., 2009*; *Dean et al., 2012*; *Carter et al., 2016*; *Evans et al., 2016*). Here, through combining observational,

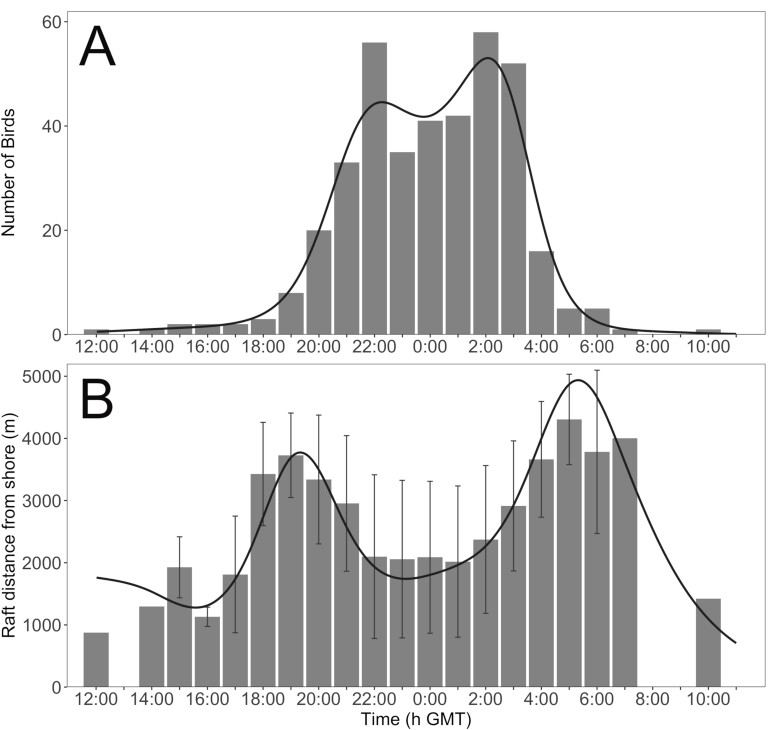

**Figure 4** The (A) number of birds and (B) mean rafting distance from Skomer Island's shoreline within 5,000 m of the North Haven colony through time across 45 days. Error bars represent standard deviation. Black lines are the fitted GAMM trends.

GPS tracking, and wind condition data, we find responses of Manx shearwater rafting locations and size to time of day and wind speed, but little support for a strong role of wind direction on rafting location.

Wind direction only has a weak effect on the rafting location of Manx shearwaters around Skomer Island suggesting that birds do not collectively seek to raft on calm waters. It is possible that rafting location is instead driven by surface currents, as is observed for streaked shearwaters (*Calonectris leucomelas*) and Scopoli's shearwaters (*Calonectris diomedea*) rafting offshore (*Yoda, Shiomi & Sato, 2014*; *Miyazawa et al., 2015*; *Sánchez-Román et al., 2019*).

Alternatively, sub-colony location may be an important factor determining rafting location because GPS tracked individuals are typically located adjacent to their breeding colony regardless of wind conditions. In this study, GPS tracked birds were all breeding at the sub-colony above North Haven where night-time rafting locations predominated in the sample (Fig. 3D). This finding aligns with *Wilson et al. (2008)*, who identified that breeding sub-colony location loosely influenced the rafting locations of Manx shearwaters around three UK islands at night.

We find that raft size marginally increases on windier days. We can only speculate on the driver of this pattern. Manx shearwaters nesting on Skomer Island spend a significantly greater proportion of time travelling than other Manx shearwaters nesting on islands near

their feeding grounds (*Dean et al., 2012*). Assuming similar wind speeds are experienced on return journeys, the increased wind speeds could reduce flight return time to the colony. The result is a high concentration of Manx shearwaters in the nearshore waters of the colony island. It is also possible that colony attendance could be cyclical, as seen in the Atlantic puffin (*Fratercula arctica*), where a large number of individuals are present for several days, followed by several days of absence from the colony (*Calvert & Robertson, 2002*). If Manx shearwater colony attendance is coordinated and non-random, it could contribute considerably to variations in raft size.

By contrast, we observe a strong shift in the distribution of rafting birds between the day and night. Before sunset, a halo effect, the absence of rafts within 300 to 500 m of the shoreline, is present around Skomer Island, but disappears during the night as more individuals return from foraging trips and as birds raft towards their colony. The halo pattern reforms before sunrise as rafts move away from the colony and leave for foraging. Observing rafts closer to the island as night progresses is concomitant with previous findings around Skomer, and two other islands in the UK, and is speculated to be associated with assessing predation threat (*Wilson et al., 2008*). However, to our knowledge, never before has a halo effect resulting from rafting behaviour been described for seabirds. Manx shearwaters have been anecdotally observed to assemble between 1 and 10 km from Skomer Island (*Brooke, 1990*), and identified to raft short distances away from the island's shoreline after nightfall (*Wilson et al., 2008*). It is well known that many nocturnal tubenose seabirds return to their burrows once light levels fall below a certain threshold, and leave before the sun rises to avoid diurnal avian predators (*Keitt, Tershy & Croll, 2004*; *Miles et al., 2013*). The halo effect may therefore represent a predator avoidance strategy. We speculate that groups of birds choose a threshold which is a safe distance from land-associated predators during the day, and accumulating closer to the colony at night could allow them to assess predation threat and weather conditions at the colony and make landfall in large groups, thereby reducing predation risk.

Aggregating in large groups at safe distances from predators provides opportunities for the birds to engage in other important activities. We find that Manx shearwaters raft for prolonged periods before and after visiting the colony which we suggest is associated with preening feathers. Individuals are likely motivated to clean themselves following burrow visits and foraging trips to maintain their plumage, as is observed for the Northern gannet (*Thiebault et al., 2014*; *Carter et al., 2016*).

## CONCLUSIONS

Thus, through combining multiple data sources, we expand our understanding of Manx shearwater habitat use and give insight into the drivers of rafting behaviour and locations around Skomer Island. Overall we find that the responses of rafting to wind conditions are variable, at least within the range of conditions presented during our studies, and may not be as ecologically significant for rafting as previously suspected (*Wilson et al., 2008*; *Carter et al., 2016*). However, we find that day-night cycles of rafting distributions are prominent for Manx shearwaters around Skomer Island and might provide a way of

waiting for dusk that reduces predation risk. If this were the case, we might expect, as we present here, changes in raft location based on the activity of diurnal predators. Based on our findings, we suggest that rafts likely provide adaptive benefits through increased predation-risk assessment and reduced personal predation risk around their breeding colony. Our findings have important implications for marine spatial planning and should be considered when making changes to Special Protection Area (SPA) boundaries (*Lennox et al., 2019*). Identifying key spatial and temporal changes of Manx shearwater rafting distributions offers insights for management agencies to make more informed decisions about how to use waters adjacent to Skomer Island during the breeding season, such as managing shipping routes to avoid conflict with rafting birds.

## ACKNOWLEDGEMENTS

We would like to thank the Wildlife Trust of South and West Wales, in particular Skomer Wardens, Bee Büche and Ed Stubbings, for their input during the conception and data collection in the study. Mark Burton and Natural Resources Wales provided weather and tidal data. Alex Brown helped with the logistics and mathematics of the inclinometer. We also appreciate the valuable input from Tony Richards. Finally, members of the Oxford Navigation Group, Amaia Mendinueta, Joe Wynn and Sarah Bond for tirelessly collecting GPS data throughout the night. We thank the Editor and three anonymous reviewers for comments which improved the manuscript.

### Funding

Amanda E. Bates and Cerren Richards were funded under the Canada Research Chairs programme (950-231832). Oliver Padget was funded by a NERC studentship (NE/L501530/1) with the RSPB as a CASE partner. This work was further supported by the Mary Griffiths Award. The funders had no role in study design, data collection and analysis, decision to publish, or preparation of the manuscript.

### Grant Disclosures

The following grant information was disclosed by the authors:
Canada Research Chairs programme: 950-231832.
NERC studentship: NE/L501530/1.

### Competing Interests

Amanda E. Bates is an Academic Editor for PeerJ.

### Author Contributions

- Cerren Richards conceived and designed the experiments, performed the experiments, analyzed the data, contributed reagents/materials/analysis tools, prepared figures and/or tables, authored or reviewed drafts of the paper, approved the final draft.
- Oliver Padget and Tim Guilford contributed reagents/materials/analysis tools, authored or reviewed drafts of the paper, approved the final draft.

- Amanda E. Bates analyzed the data, authored or reviewed drafts of the paper, approved the final draft.

## Animal Ethics

The following information was supplied relating to ethical approvals (i.e., approving body and any reference numbers):

Both the University of Oxford Animal Welfare and Ethical Review Body and the BTO Special Methods Technical Panel gave full approval for this research and further deemed GPS tracking of Manx shearwaters to fall below A(SP)A threshold and thus no licence is required from the home office. The BTO oversees telemetry of wild birds in the UK through their Special Methods Technical Panel, who approved the GPS tracking techniques used here through the endorsements to Tim Guilford's ringing permit (Permit Class and Number: / C / 5311).

## Field Study Permissions

The following information was supplied relating to field study approvals (i.e., approving body and any reference numbers):

A bird ringing permit was obtained from the British Trust for Ornithology (Permit Class and Number: / C / 5311). Field experiments on Skomer Island were approved by the Islands Conservation Advisory Committee (ICAC).

## Data Availability

The raw code is available in Code S1. Behavioural observation and GPS datasets are available in Data S1 and S2.

Raw data has not been provided since data processing included stages whereby data were manually edited without using code. These stages have been highlighted in Code S1. We have provided the final processed data on which the statistical analyses were performed.

## Supplemental Information

Supplemental information for this article can be found online at http://dx.doi.org/10.7717/peerj.7863#supplemental-information.

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
