# Peer review of "Manx shearwater (Puffinus puffinus) rafting behaviour revealed by GPS tracking and behavioural observations"

_PeerJ, doi:10.7717/peerj.7863_

## Round 0.1 · original submission · Minor Revisions

The reviews were quite favorable and I agree that only minor revisions are needed. I have a few questions, comments, and editorial suggestions.

Comments on predictions:
Lines 108-109: Don’t procellariiform seabirds prefer to fly in windy conditions? Maybe a hump-shaped relationship with more rafting in low winds and very high winds?

Methods and results clarifications:
L 141: Why times H? Wouldn’t that reduce the height (because H < 1) rather than adding to it?
What is D? Please explain this better. This seems like a basic trigonometry problem: Raft distance = total height above water surface * tan(Ɵ).
L 199: Please specify whether this is using observations, GPS locations, or both.
L 203: Autocorrelation in raft size, wind speed, or both?
L 213-215: This confused me at first. After I read the results, I got it, but a clearer description here would help.
L 239-241: Would a nonlinear function fit better?
L 241-242: Was there autocorrelation? Say something about it?

Editorial comments:
Line 65: Add a comma, “typically on isolated land masses, during the breeding season”.
L 95: “shearwater” (not plural).
L 96: “procellariiform”. Don’t believe everything Microsoft spell checker tells you.
L 96: “With more than 316,000”.
L 97: “shearwaters”.
L 107: “We predicted that seabirds would shelter” OR “We predict that seabirds shelter”.
L 115: This is awkward. Maybe “Field methods: behavioural observations”.
L 120: Reference Figure 1 after the list of observation stations.
L 133: Is Wooltack Point on Skomer Island? If so, add to Figure 1. If not, but it’s on Figure 2, add it there. If it doesn’t fit on either figure, say how far away it is.
L 147: Same as Wooltack Point. If it fits on either map figure, include it. Otherwise say how far away it is.
L 163: “returned”, not “retuned”.
L 272-273: When you say “shearwaters” you imply either your study species or all species of shearwater. Clarify this a little.
L 308: “Procellariidae”.

Figures and Tables:
Figure 1: Add at least on more circle size to the legend (maybe corresponding the smallest circle in the data).
Add symbols for the sampling location.
Add the tide station and weather station if they are on the island.
Figure 2: Say something about the lines of dots. Do they represent flying or rafting behavior?
Add the tide station and weather station if appropriate.
Table 1: Explain the Thetas.

Reviewer 1 ·

Basic reporting

I generally found the article very well written and easy to read. I have some thoughts about the structure which I describe in my general comments.

Experimental design

I feel the research questions about environmental effects could be better linked to some of the ideas the authors mention about the purposes of rafts to assist predator avoidance, socialising/preening or acting as compasses indicating the direction of foraging patches.

Validity of the findings

The methods and statistics seem to be extremely robust.

As mentioned above, In terms of interpreting the findings I would like to generally like to see a link between the environmental variables such as wind, and the other speculation about the purpose of rafts. Currently these ideas seem somewhat disconnected.

Additional comments

I was extremely interested to read this MS studying the factors influencing rafting behaviour of shearwaters. The authors have collected an impressive amount of GPS data and combined it with behavioural observations to answer questions about how environmental variables affect rafting behaviour. The manuscript is clearly written and easy to read. However, I feel the paper would benefit from some links to broader questions about group behaviour. This could be achieved in part by better integrating the ideas about the anti-predation benefits of rafting throughout the paper.

While I found it very well written, I am not entirely sure about the current structure of the introduction. I would prefer to see a more general introduction. I feel that a slightly bolstered second paragraph possibly with some basic discussion about group behaviour in other taxa, potential adaptive benefits, trade-offs would make for a stronger start. The first paragraph could be repurposed elsewhere to demonstrate why seabirds are good to examine these types of questions.

Similarly, I would then like to see the linking of the potential environmental variables back to larger questions about the adaptive significance of these rafts. For example, if rafting behaviour is indicative of the direction that individuals will depart to or have returned from foraging patches, how might this be affected by wind? Would individuals be departing or returning downwind, causing changes to the distribution? How might this interact with the authors ideas about predation risk? Etc.

I think the hypotheses about predators should also be introduced earlier and perhaps developed a bit more, it feels a little last-minute in the introduction at the moment.

The experimental and statistical methods are very clearly presented. However I feel a little more detail on the partitioning of GPS tracks between rafting and flight would be beneficial.

Details on the number of maximum number of GPS devices simultaneously deployed would be useful to include in the results.

My comments on the discussion are similar to those about the introduction. While it is well written, I would like to see links to some broader speculation about the adaptive significance of rafting behaviour. This could be achieved by expanding on some of the ideas about the purpose of the raft being for socialising/preening or predator avoidance.

Reviewer 2 ·

Basic reporting

This manuscript provides valuable new knowledge about rafting behaviour on seabirds by combining observational and GPS-tracking on Manx shearwaters, as well as wind conditions. Authors conclude size of the rafts and their location is influenced by wind speed and time of the day, although wind direction does not seem to play a role on rafting locations.
The manuscript is very well written, in an unambiguous and professional English. It is also well structured and main ideas are clear and well explained.
Introduction is clear and well structured, although there are some things I think that can be improved.
First paragraph is too general to me. Taking into account this manuscript has a clear focus regarding rafting behaviour, I would start talking about rafting in the first paragraph (now you have to wait until paragraph two to see the focus of the manuscript).
Second paragraph is very good and key to understand why seabirds perform rafts. However, perhaps you could also include a brief explanation on which groups or species are known to perform rafts and which ones not.
In lines 88-89 you say “… the importance of various environmental and biotic drivers on rafting behaviour…” Since this is the focus of the manuscript, I would better describe which are these environmental and biotic drivers that have been suggested to affect rafting behaviour. I would briefly explain them and then later focus on the wind, which is the one you actually look at. I would also better explain why you think wind might be key for your species, to justify why you did look at wind and not at other environmental or biotic drivers.
Both in the objectives and particularly in the discussion you say you used “environmental data” to understand rafting behaviour. However, to me, if you only look at wind, I am not sure you can claim to work with “environmental data” in a general sense, because it seems you worked with other environmental factors also affecting rafting behaviour. I would say “wind conditions” instead or something like this.
In the objectives, you talk about whether raft distance from shore changes through time. However, nowhere in the introduction you have explained this hypothesis. I would include this idea in the introduction so that it fits with the objectives. Indeed, it is a very interesting result!

Experimental design

The experimental design of the manuscript is well explained and with well defined and supported questions. Authors identify the gaps of knowledge they are going to fill in in a clear and ordered way. Ethics on the manuscript are correct.
Methods are detailed, clear and well explained, although I have a few comments. In the models with GPS data, did you take into account the individual? I do not know if it may play a role, but it can be included as a random factor in models. Also, did you think about including the moonlight intensity as a driver? I do not know if it can be important, but as it is important regarding the entrance of birds to the colony, it may also play a role with rafting behaviour.
L 130-132: I would simply say “Rafts observed at the 2000 m boundary and beyond were excluded in our analyses because…”
L178: So you only applied a “land-filter”? Didn't you do another filter (for instance velocity) to remove erroneous locations?
L184: “… Manx shearwaters flight speeds…” so you only used velocity to define rafting locations? Cannot birds also be intensively foraging with very low speed? What about the turning angle in these locations? I am thinking about EMBC approach (Garriga, J., Palmer, J. R., Oltra, A., & Bartumeus, F. (2016). Expectation-maximization binary clustering for behavioural annotation. PLoS One, 11(3), e0151984.).

Validity of the findings

Results and discussion are detailed, well explained and well supported by statistical tests and figures. I have a few comments.
L231: “around Skomer Island” you mean in the radius you looked at? Or in general?
L289-293: I am sorry, but I do not fully understand the relationship with the dual foraging strategy and the size and number of rafts. Could you explain it in another way?
Conclusions are very good, but they go beyond what it is explained in the introduction. To make the manuscript “circular” it would be a good idea to also talk about the importance of understanding rafting behaviour in conservation in the introduction. In this way conclusions will match with the introduction.
Fig. 1. I would mark in the island where exactly the observational points were. I would also include that this figure comes from observational data, and that observations were made in the 2 hours before sunset.
Fig. 2. I would include the number of birds in the figure caption. Also, I do not know if it is necessary to have colour in this figure (you can make the star black with white contour or something like this and you do not need to use the colour).

Reviewer 3 ·

Basic reporting

Highly professional structure and clarity throughout.
Clear logical development of aims that are explored in the study.
Referencing is appropriate, but could be improved in places: sometimes a single, species-specific source is used to support the wider value/evidence base for an approach or concept: more general reference might be added (e.g. a review to complement a specific paper). Example is L68 Dean et al (2012), maybe complemented by broader scope or sources.
L249 Can the nature/directionality of the effect of wind be clarified here? If some benefit to land shelter or exposure is present, pls add specific detail.
Fig 2 demonstrates the 'halo' effect, but how the tracks provide evidence for this might be clarified in the body text or the figure legend... that the tracks are further from land by day and closer by night may not be so obvious to the reader

Experimental design

The study is well-designed, research questions clearly articulated and addressed.
High standard of technical expertise and ethics.
The knowledge gap is clearly identified.
Methods appropriately reported.
Is the inclinometer approach novel, or inspired by uses in other fields? Some clarification here would be valuable.

Validity of the findings

Clear rationale for the study and interpretation of results provided.
Elegant and concise.

Additional comments

I enjoyed reading this paper, thank you for the opportunity to read it. I liked the combination of approaches of tracking data and the more 'old school' inclinometer, the neat logical framework and the clear, concise writing.

---

## Round 0.2 · accepted · Accept

Please note that I am passing 2 minor editorial changes along to the production staff. Line 49: "... and cleaning." is somewhat ambiguous. Do you mean avoiding cleaning? Cleaning what? I suggest replacing "and cleaning" with "and for cleaning feathers" to make this clear. Line 207: the correct term is geodesic or geodetic.